# Linking Social Media Marketing Activities to Revisit Intention through Brand Trust and Brand Loyalty on the Coffee Shop Facebook Pages: Exploring Sequential Mediation Mechanism

Blend Ibrahim [1,*] , Ahmad Aljarah [2] and Dima Sawaftah [2]

1 Department of Tourism, School of Tourism and Hospitality Management, Girne American University, Girne 99300, North Cyprus, Mersin 10, Turkey

2 Marketing Department, Faculty of Business and Economic, Girne American University, Girne 99300, North Cyprus, Mersin 10, Turkey; ahmadaljarah@gau.edu.tr (A.A.); deemamubaslat@gmail.com (D.S.)

* Correspondence: blendibrahim@gau.edu.tr

**Abstract:** Social media marketing (SMM) is a new field that involves the marketing of goods, services, information, and ideas via online networks and social media. Drawing on the stimulus-organism-response framework, this study aims to examine how social-media-marketing activities (SMMA) affect brand loyalty, brand trust, and revisit intention (returning to the same place in the future) for coffee shops in Northern Cyprus. Empirical evidence was collected from 415 undergraduate students who follow specific coffee shops on Facebook, and a structural equation modeling approach was applied. The results showed a significant positive influence of SMMA on brand loyalty, brand trust, and revisit intention. The findings show that SMMA are a stronger predictor of revisit intention than brand loyalty and brand trust. Furthermore, brand loyalty and brand trust are significant mediators in the relationship between SMMA and revisit intention. Additionally, the sequential mediation effects of brand loyalty and brand trust in the relationship between SMMA and revisit intention are supported. Overall, with effective SMMA from coffee shops on Facebook, the customer grows confidence in the brand, which increases the level of brand loyalty. This, in turn, encourages revisit intention of the customer. As a result, brand executives on social media platforms (in this case, Facebook) should promote specific SMMA for their brands and engage in such activities to creates brand trust and brand loyalty. These findings contribute to the literature by examining the relationship between SMMA and revisit intention and exploring how SMMA affect revisit intention by adding brand loyalty and brand trust as mediators.

**Keywords:** social media marketing activities; brand loyalty; revisit intention; brand trust; customer behavior; coffee shops; hospitality industry; North Cyprus

## 1. Introduction

Social media power has shifted the actions of consumers and companies. For example, to get breaking news, people have started using Twitter instead of newspapers or television. Even YouTube celebrities are now more famous than Hollywood stars [1]. Businesses now create social media fan pages for their brands through Facebook, Instagram, and Twitter to reach many consumers, which has led to new types of marketing plans, called social media marketing (SMM). With different communication tools and content in the tourism and hospitality industry, SMM appeals to researchers and specialists for many reasons, such as the advantages that online social media networks offer as hospitality business marketing tools. SMM is described as the method of using social media networks to create, interactively deliver, and turn corporate contributions beneficial to corporate stakeholders [2]. Hospitality businesses should apply SMM to make a substantial profit-efficient investment [3] and provide a perfect platform for hotels to discover their customers' views [4]. Because half of the world's population uses social media networks [5], there are

many opportunities to expand on various social media platforms and incorporate this trend into any business's marketing plan in the hospitality industry. Different types of tourism and hospitality businesses, such as hotels, luxury hotels, travel agencies, restaurants, and coffee shops, benefit from developing social media platforms content, easy interaction with customers, and an interconnection between different types of social media platforms. In addition, from the same point of view as online communities, Kim and Ko [6,7] created new social media marketing activities (SMMA) in some areas (entertainment, interaction, trendiness, customization, and word of mouth (WOM)) to discover how SMMA increase consumer equity and purchase intention in terms of a luxury fashion brand. SMMA refer to interactions and communication between consumers and brands [8]. SMMA can amplify and satisfy the differing needs of individuals, as well as enhance the appearance of a consumer–brand relationship [9].

SMMA examined various behavioral outcomes, including positive individual behaviors and attitudes [10–12]. The most important individual outcomes for consumers highlighted in this research process are revisit intention and brand trust [13]. Revisit intention is described as the readiness or willingness of a visitor to revisit the same goal [14], while brand trust is "the willingness of the average consumer to rely on the ability of the brand to perform its stated function" [15] (p. 82).

While the effects of SMMA on branding and brand equity have been investigated by some studies [10,12,16], few researchers have explained how and why SMMA affect revisit intention [17], brand loyalty [18,19], and brand trust [20,21].

The relationship between SMMA and behavior intention (purchase intention and revisit intention) is well researched in literature. Although most SMMA studies of these types of relationships have explored how SMMA unconditionally and directly affect behavior intention [11,22,23], they do not focus on the factors that explain the connection between SMMA and behavior intention. While these concepts may be imperative to business success on social media platforms, these SMM-efforts boundary conditions remain less clear. We explore the essence of the relationship between these constructs in the tourism and hospitality industry (coffee shops) empirically [20,21]. Thus, this research explores the literature void by analyzing SMMA and behavioral responses such as revisit intention, brand loyalty, and brand trust.

This first gap in the literature paints an incomplete picture of the SMMA–customer-responses relationship and limits our understanding [9,12,16,24]. Our research is one of the few studies that examines the purpose, brand trust, brand loyalty, revisit intention, and SMMA together in the tourism and hospitality industry (specifically, coffee shops).

A second gap in the literature concerns the underlying mechanism of how SMMA influence the tourism and hospitality business and the customers who follow the Facebook pages of coffee shops. Most studies have focused on the relationship between SMMA and consumer responses in the luxury industry [6,12,25–27], education on Facebook pages [28], smartphone users [11], the airline industry [20,29], and the service industry [30]. These shortcomings for the tourism and hospitality industries match the call by recent research studies to examine SMMA in a wider variety of industries and countries (i.e., developed, emerging, or developing) [9,10,12,21,23,24].

This study selected global franchise coffee-shop brands in North Cyprus, like Gloria Jean's Coffees and Caffè Pascucci. With 14 coffee shops located in the three main cities in North Cyprus, both brands are the largest coffee brands in the country (Kyrenia, Nicosia, and Famagusta). In addition, Gloria Jean's Coffees ranks third globally among first-class coffee brands in the world and offers the customers different kinds of products, including hot and cold coffee drinks, juice and tea collections, and dessert foods [31]. The global coffee brand is dynamically involved in social media platforms through an active newsfeed, promotions, photos about products and services, offers to the customers to generate content related to the products, and discussion forums to express product criticism or recommendations [32]. The coffee-shop businesses that embrace social media can also advance customer-service reputations, gather more positive mentions, and motivate

conversation among users [33]. Lastly, the overwhelming majority of recent empirical studies on SMMA have been conducted in Asia [11,16,20]. Therefore, there is a need to investigate SMMA in different cultural settings like Northern Cyprus [12,21,23] or in a different context like coffee shops [10,17,24]. Therefore, the current study fills this research gap by empirically examining SMMA's role in promoting brand trust, brand loyalty, and revisit intention in the top two franchise coffee companies in Northern Cyprus.

The third research gap concerns the conditions under which SMMA might enhance revisit intention in the hospitality context (specifically coffee shops). No current research examines the mediating roles of other aspects of cognitive and emotional states such as brand trust and brand loyalty in the relationship between SMMA and revisit intention [20–22]. By examining the relationship between SMMA and revisit intention considering a brand trust and brand loyalty as a mediating factor, this research bridges this gap in the SMMA literature and calls for further investigation into different mediation roles between SMMA and consumer responses.

The fourth research gap covers the sequential mediation effects of brand loyalty and brand trust in the relationship between SMMA and revisit intention. SMMA intend to influence revisit intention through the mediation effect of brand trust and brand loyalty. Therefore, if the relationship between the study independent and dependent constructs were complex, a change in one mediator might lead to a change in another mediator that must investigate the sequential mediation relationships as a superior approach to understanding potential connections between all variables [34]. Therefore, by considering a sequential mediation effect, this relationship between customer and company indicates the possible existence of a mediating effect of brand trust and brand loyalty between SMMA and revisit intention. This effect has not yet been investigated. We aim comprehensively to establish how SMMA is linked to the customer revisit intention through brand loyalty and brand trust.

This work contributes to SMMA literature in the tourism and hospitality industry in several ways. The study first examines SMMA's impact on brand trust, brand loyalty, and revisit intention, which extends SMMA research in the tourism and hospitality industry. Second, it analyzes processes concerning how brand trust mediates the relationship between SMMA and revisit intention or how brand loyalty mediates the effects between SMMA and revisit intention. Third, we extend this previous literature work by researching the service-industry relationship between brands and customers. The samples are coffee-shop customers in Northern Cyprus interested in social media platforms (specifically Facebook pages for coffee shops). Fourth, a sequential mediation model is developed and tested to demonstrate how SMMA affect brand trust in revisiting intention toward brand loyalty.

## 2. Theoretical Background

*The S-O-R Theory-Based Research Model*

The S-O-R model (stimulus-organism-response) is useful for supporting an integrative model in this study [35]. In the context of SMMA, the S-O-R model discovered in the study of Koay et al. [10] examined the relationship among stimuli (S) (i.e., SMMA), organism (O) (i.e., brand experience), and response (R) (i.e., customer brand equity). The S-O-R model states that specific features of external environment factors or stimuli (here, SMMA) and provokes the cognitive and emotional internal state as an organism (here, brand trust and brand loyalty) of consumers, which produces some behavioral response (here, revisit intention). The term "organism" refers to the inner states of perceptions, feelings, and thinking exercises [36]. In the context of tourism and hospitality, the S-O-R model states that specific features of an environment or stimuli (i.e., corporate social responsibility) lead toward a certain inner state of the consumer or organism (i.e., benevolence trust) and response (i.e., customer citizenship behavior) [37].

Figure 1 demonstrates the relations among the study variables: SMMA dimensions, brand loyalty, and revisit intention.

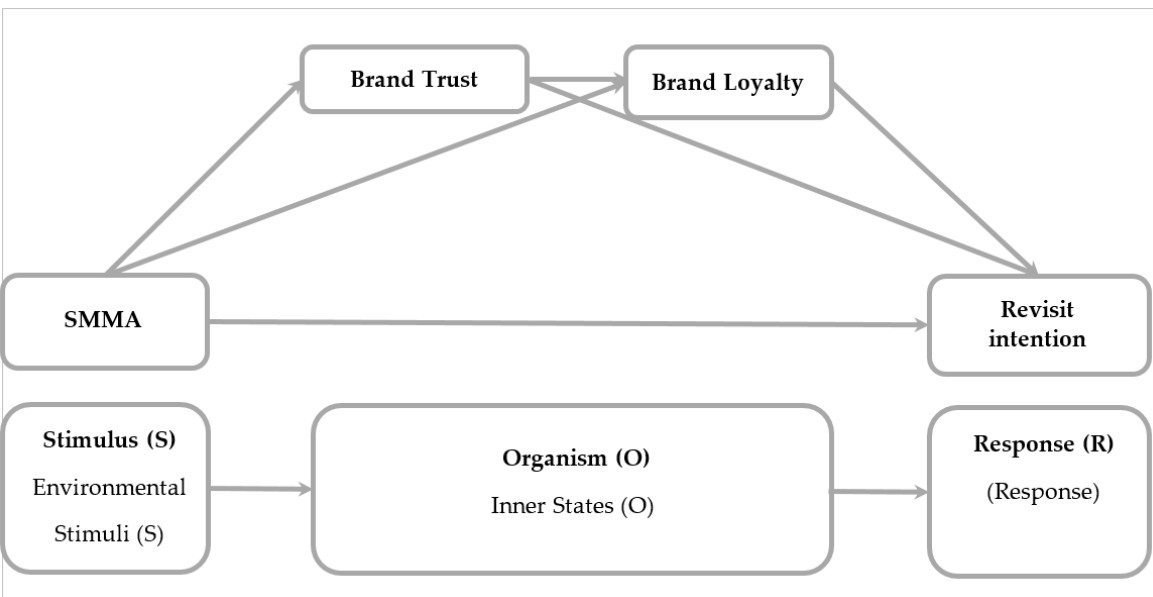

**Figure 1.** The S-O-R theory-based research model.

## 3. Hypothesis Development

### 3.1. Social Media Marketing Activities and Revisit Intention

Since prior works on SMMA, an increasing research stream on this area has been established [6,7]. For luxury brands, SMMA refer to the active marketing communication approaches related to customers that are influential for the brand and encompasses five dimensions: entertainment, interaction, trendiness, customization, and word-of-mouth (WOM) [7]. A recent meta-analysis article viewed SMMA as "promotional and relational communication tools that complement organizational marketing strategies application by offering enhanced interactivity through online relationships between organizations and consumers" [9] (p. 5). Alternatively, SMMA can defined as a "considerable role in building trust and the consumer–brand relationship, which lead to positive business outcomes in turn" [38] (p. 698). SMMA, as a promotional communication tool, support traditional marketing efforts in Facebook coffee-shop pages.

From another side, the idea of revisit intention stems from behavioral intention, which is identified with various meanings (e.g., repurchase, intention to buy, and intention to revisit) [39]. Satisfied travelers are more inspired to help and revisit the same place visited by others [40,41]. Earlier studies examined revisit intention in the tourism industry and social media platforms. The study of Ruiz et al. [42] examined social media users regarding Spain's tourism context and suggested that the influence of emotion on postpurchase behavior is more important for active social media users than for passive users. In the context of online repurchase, the customer intended to buy or was interested in buying a certain product or service and inspired the user to visit or revisit the same venue [43,44]. Also, users or customers with high levels of involvement in a hotel Facebook page had a higher intention of revisiting the hotel social media pages [45]. There is little empirical research on the link between SMMA and revisit intention in the tourism context. For instance, the investigation of Ibrahim and Aljarah [17] in the relationship between SMMA and revisit intention of five-star hotels in Northern Cyprus found that SMMA can improve revisit intention in the tourism industry. Jung, Kim, and Kim [46] reveal that consumer attitudes toward the online brand community positively promote their perception about visiting this website again. Zaenab et al. [47] reveal that electronic WOM toward the online brand community positively affect revisit intention for travelers postearthquake in Indonesia.

Additionally, SMMA enhances and reinforces the consumer's interaction with a hotel or brand. The business corporation tries to influence consumer attitude to be positive and

active to repurchase or revisit [48,49]. For that reason, it is reasonable to believe that SMMA have a positive effect on revisit intention [17]

Therefore, it is hypothesized:

**Hypothesis 1 (H1).** *Social media marketing activities have a positive effect on revisit intention.*

### 3.2. Social Media Marketing Activities and Brand Loyalty

Brand loyalty can be described as an attachment to a particular company and its products [50]. Further, brand loyalty is "a deeply held commitment to re-buy or re-patronize preferred product/services consistently in the future" [51]. Different measures conceptualize it: behavioral (e.g., [15,52]) and attitudinal (e.g., [53,54]). Marketing studies show that behavioral loyalty is a repurchase process that attracts and creates customer loyalty to a particular service provider [52]. Attitudinal loyalty is dissimilar to behavioral loyalty, as repeated purchases characterize it through the customer's mental and emotional states, which then mediate stimuli and responsiveness.

Some studies have established a direct link between SMMA and brand loyalty in marketing literature. For instance, Ebrahim [19] indicated that SMMA directly influence brand loyalty and indirectly influence brand equity mediated by brand trust. The study was applied to 287 users who followed telecommunications companies on social media. Wibowo et al. [21] revealed that SMMA and customer experience positively affect customer relationship quality and customer behavioral outcomes, based upon a sample of users who have experience with social media online shops. Effective communication between consumers and brands or stronger two-way communication interactivity between customers and companies about products and services offer a compelling potential relationship concerning brand building [55]. Consequently, the more effective the interaction and connection is, the stronger the consumer–brand relationship will be, and the more loyal the consumers will be to the brand.

Therefore, it is hypothesized that social media marketing activities will contribute to brand loyalty:

**Hypothesis 2 (H2).** *SMMA have a positive effect on brand loyalty.*

### 3.3. The Mediating Role of Brand Loyalty

We conceptualize SMMA as utilizing a mediated (indirect) effect on revisit intention via brand loyalty. The study of Cheung et al. [11] asserted the role of SMMA in building value cocreation and customer–brand engagement in smartphone brands, as well as repurchase intention and ongoing searches in China and Hong Kong. However, the previous scholars did not come across any studies clarifying the mediating effects of brand loyalty in explaining the relationship between SMMA and revisit intention. Consequently, social media's strong consumer–brand relationship led to brand loyalty [56]. In addition, Ismail [57] revealed that SMMA are an active tool that strengthens relationships with consumers and builds brand loyalty within the social media-based brand community. Some studies in the tourism context [17,58] stated that customer loyalty increases revisit intention. Furthermore, some studies recognize that revisit intention is at the heart of loyalty [59]. Building on the arguments presented in the existing literature, we posit the following:

**Hypothesis 3 (H3).** *Brand loyalty mediates the relationship between SMMA and revisit intention.*

### 3.4. Social Media Marketing Activities and Brand Trust

SMMA convert consumers into marketers and promoters who produce, manage, and share online information related to firms, products, and services [9,57]. While some studies in different industries have investigated SMMA and trust [18,20,60], we argue that SMMA on a coffee shop's Facebook page will enhance brand trust. Thus, the social interaction and connection between consumers and firms influence the consumers' brand trust [61,62] and

produce social support over social media to impact that trust [63]. Trust is considered one of the key factors to an enhanced relationship between a consumer and a certain brand [6], which matches with the function of SMMA to provide better communication for marketing departments or firms and build long-term relationships with customers [18,64]. We are, therefore, proposing the following:

**Hypothesis 4 (H4).** *Social media marketing activities have a positive effect on brand trust.*

### 3.5. The Mediating Role of Brand Trust

We conceptualize SMMA as utilizing a mediated (indirect) effect on revisit intention via brand trust. SMMA are supportive activities with strong interaction and connection between consumers and firms [61,62]. Relationship marketing reveals that influences of behavior on outcomes are generally mediated via satisfaction, commitment, and trust [65]. Thus, we expect a brand trust impacts revisit intention. Lin and Lu [66] proposed that trust substantially influences repurchase intention, while Chiu [67] found that consumer trust stages can affect repurchase intention or revisit intention. SMMA may indirectly influence revisit intention via the mediating effect of brand trust.

We, therefore, hypothesize the following:

**Hypothesis 5 (H5).** *Brand trust will mediate the relationship between social media marketing and revisit intention.*

### 3.6. The Serial Multiple Mediating Effects of Brand Trust and Brand Loyalty

So far, this theoretical study suggests that SMMA in the social media community encourage the creation of brand trust, which, in turn, increases brand loyalty, eventually leading to greater revisit intention. Furthermore, Chaudhuri and Holbrook [15] observed that customers with favorable and confident emotions were strongly linked to a high brand loyalty level. Thus, customers can generate trust and develop passionate links to brands. In other words, "building relationships on the social Web is practically a guaranteed way of deepening brand loyalty" [68] (p. 170). Brand trust leads to brand loyalty or commitment because trust generates highly valued exchange relationships [69]. Hence, SMMA increase brand trust, which affects brand loyalty and, consequently, improves revisit intention. Because this relationship between brand loyalty and revisit intention has been confirmed by some studies [70,71], we desire to propose and examine the sequential mediation chain to precisely recognize how SMMA affect revisit intention. It is expected that brand trust and brand loyalty will act as serial mediators in the link between SMMA and revisit intention.

Based on the conclusions above, we hypothesize the following:

**Hypothesis 6 (H6).** *The relationship between SMMA and revisit intention is sequentially mediated by brand trust and brand loyalty.*

## 4. The Methodology of the Study

### 4.1. Data Collection and Participant Characteristics

A self-administered survey was distributed to collect and examine the conceptual research model. A convenience sample method was employed to collect data from university students in North Cyprus with motivation to receive extra credit. University students were the preferred sample of study for several reasons: students are considered heavy users of social media [72]; students spend a significant portion of their waking hours daily with technology [73]; the majority of coffee-shop customers are students [74,75]; the university student population in North Cyprus is considered to be 30 percent, with more than 100,000 [76] from the total number of 350,000 citizens [77].

This study selected two coffee shops (Gloria Jean's Coffees and Caffè Pascucci) that are international brands and are well known as places for studying. The Facebook followers of the two mentioned coffee shops have been used as a stimulus in this study. To exclude the

responses of students who were not following either coffee shops, screening questions were added at the beginning of the survey, asking if the participant following any coffee shop investigated in this study. During class time, 502 online-prepared surveys were distributed, and 415 valid questionnaires were collected.

*4.2. Measurement Scales*

Multi-items were measured with a five-point Likert scale (1 = strongly disagree, 5 = strongly agree) to estimate the study variables. All measures were adapted from literature and modified to suit the study context. SMMA were measured by adopting 10 items from Kim and Ko [7]. A four-item scale measuring brand trust was taken from Laroche et al. [78] and Chaudhuri and Holbrook [15]. To measure brand loyalty, we employed [79] a measurement instrument. Finally, four-item scales measured revisit intention [80,81].

## 5. Results

*5.1. Sample Characteristics*

The majority of students were male (67%), while females accounted for only (33.2%). The dominant group aged between 18 and 25 years old (80.6%), and 19.4% of participants aged higher than 25 years old. Sixty eight of the participants were followers of Gloria Jean's Coffees' Facebook page, while the rest were followers of Caffè Pascucci Facebook page.

*5.2. Validity and Reliability Analysis*

This study employed the statistical software IBM SPSS 23.00 and IBM SPSS Amos 24.00 to conduct the statistical analyses. Confirmatory-factor analysis using maximum-likelihood estimation was assessed to examine the instrument, convergent, and discriminant validity. The instrument validity was examined by using the goodness-of-fit model [82]. The result revealed that the data collected fit well with the research conceptual model ($x2 = 1.79 < 3.00$; GFI = 0.93 > 0.90; CFI = 0.97 > 0.90; NFI = 0.94 > 0.90; RMSEA = 0.04 < 0.08; and PCLOSE = 0.92 > 0.05). The average variance extracted (AVE) was used to examine convergent validity. As shown in Table 1, the AVE values were higher than 0.5, exceeding conventional threshold levels suggested by Fornell and Larcker [83]. Furthermore, each construct's factor loadings were statistically significant and higher than 0.65. Thus, the research measurements had a high-convergent validity level [82,83]. Discriminant validity was established to ensure that items were strongly correlated with their indicators. The results in Table 2 show the high level of discriminant validity, as the square root of the average variance of extracted values for each construct exceeded the correlation coefficients between the constructs [82,83]. Cronbach's alpha ranged between 0.86 and 0.92, indicating adequate internal-consistency reliability, as it is exceeded a conventional threshold (>0.7) [84]. Thus, the measurement model was confirmed to be reliable and valid.

**Table 1.** Summary of the measurement model.

| Latent Constructs | λ |
|---|---|
| **Social media marketing activities ($\alpha$ = 0.91; CR = 0.90; AVE = 0.50)** | |
| Using coffee shop brand X social media is fun. | 0.65 |
| The content of coffee shop brand X social media seems interesting. | 0.71 |
| Coffee shop X social media enables information sharing with others. | 0.65 |
| Conversation or opinion exchange with others is possible through coffee shop X social media. | 0.65 |
| The content shown on coffee shop X social media is the newest information. | 0.7 |
| Using coffee shop X social media is very trendy. | 0.73 |
| Coffee shop X social media offers a customized information search. | 0.78 |
| Coffee shop X social media provides customized service. | 0.74 |

| Latent Constructs | λ |
|---|---|
| I would like to pass along information about the brand, product, or services from coffee shop X social media to my friends. | 0.78 |
| I would like to upload content from coffee shop X social media to my blog or microblog. | 0.66 |
| **Brand trust for coffee shop X social media ($\alpha$ = 0.89; CR = 0.89; AVE = 0.68)** | |
| Coffee shop X brand gives me everything that I expect out of the product. | 0.75 |
| Coffee shop X brand never disappoints me. | 0.85 |
| I rely on coffee shop X brand. | 0.85 |
| Coffee shop X has honest brand. | 0.85 |
| **Revisit intention for coffee shop X social media ($\alpha$ = 0.86; CR = 0.87; AVE = 0.62)** | |
| I want to revisit this coffee shop X brand in the next days. | 0.71 |
| I will probably revisit this coffee shop X brand soon. | 0.87 |
| I will definitely revisit this coffee shop X brand next year. | 0.81 |
| I will soon make a return visit to this coffee shop X brand. | 0.77 |
| **Brand loyalty for coffee shop X social media ($\alpha$ = 0.92; CR = 0.92; AVE = 0.74)** | |
| I would rather stick with a coffee shop X brand I usually visit than try something I am not very sure of. | 0.81 |
| I have certain types of coffee shop X brand that I always visit. | 0.86 |
| I consider myself to be loyal to one coffee shop X brand. | 0.91 |
| I have confidence in a coffee shop X brand that I always visit. | 0.87 |

**Table 2.** Assessing discriminant validity.

| Constructs | BL | RI | BT | SMMA |
|---|---|---|---|---|
| Brand loyalty (BL) | 0.863 | | | |
| Revisit intention (RI) | 0.440 | 0.792 | | |
| Brand trust (BT) | 0.290 | 0.560 | 0.826 | |
| SMMA | 0.300 | 0.410 | 0.570 | 0.707 |

*5.3. Hypothesis Testing*

The structural-equation-modeling method was used to examine the direct relationships in the conceptual research model. The research conceptual model reported for a good model fit ($x2$ = 1.94 < 3.00; GFI = 0.91 > 0.90; CFI = 0.95 > 0.90; NFI = 0.94 > 0.90; and RMSEA = 0.05 < 0.08). The research model could explain 28% of the variance in brand trust, 10% of the variance in brand loyalty, and 37% of the revisit intention variance (see Figure 2). We followed several steps to examine our hypotheses (see Table 3).

First, the value of the beta coefficient of the direct relationship between SMMA and revisit intention was significant at level 0.05 ($\beta$ = 0.13, $p$ < 0.05). Therefore, we accept Hypothesis 1. In the second step, we examined the mediation effect of brand trust and brand loyalty in the relationship between SMMA and revisit intention. The direct relationship of SMMA with brand trust and brand loyalty was examined. The results revealed a significant relationship between SMMA and brand trust at level 0.05 ($\beta$ = 0.71, $p$ < 0.05) and brand loyalty at 0.05 ($\beta$ = 0.24, $p$ < 0.05). The third step, the indirect relationship between SMMA and revisit intention, was examined separately and sequentially through brand trust and brand loyalty. The 95% bias-corrected bootstrapped confidence intervals (N = 5000) were estimated [85]. The findings indicated that SMMA have a significantly lower indirect impact on revisit intention through a brand trust ($\beta$ = 0.26). A bootstrapped estimate of the indirect effect was reported for a statistically significant indirect path at 95% CI [0.19, 0.33]. Henceforth, we accept Hypothesis 2. The value of the beta coefficient

of the indirect relationship between SMMA and revisit intention through brand loyalty reported a positive value (β = 0.06) and a statistically significant path at 95% CI [0.02, 0.10], supporting Hypothesis 3. Then, we examined the serial mediation effect of brand trust and loyalty on the relationship between SMMA and revisit intention. The result showed that the indirect effect of SMMA and revisit intention through brand trust, and brand loyalty was statistically significant (β = 0.03; 95% CI [0.01, 0.05]). Thus, Hypothesis 4 is strongly supported.

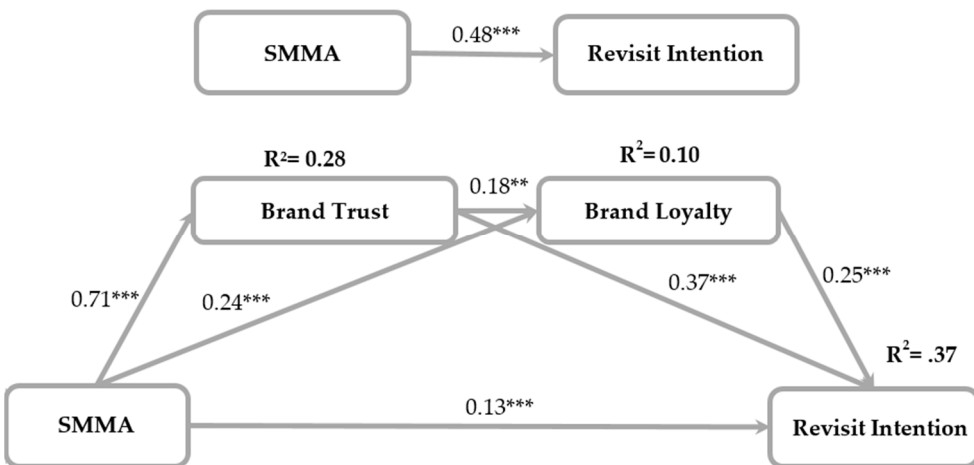

**Figure 2.** Results of hypothesis testing. Note: ** = $p < 0.01$; *** = $p < 0.001$.

**Table 3.** Empirical results.

| Paths From→ To (β) | BT | BL | RI |
|---|---|---|---|
| SMMA | 0.71 *** | 0.24 ** | 0.55 *** |
| BT | | 0.18 ** | 0.37 *** |
| BL | | | 0.25 *** |
| | B | CI low | CI high |
| **Direct effect** | | | |
| SMMA → RI | 0.13 | 0.01 | 0.24 |
| **Mediation effect** | | | |
| SMMA → BT → RI | 0.26 | 0.19 | 0.33 |
| SMMA → BL → RI | 0.06 | 0.02 | 0.10 |
| SMMA → BT → BL → RI | 0.03 | 0.01 | 0.05 |
| **Total effect** | | | |
| SMMA → RI | 0.48 | 0.37 | 0.59 |
| **R2** | | | |
| BT | 28% | | |
| BL | 10% | | |
| RI | 37% | | |

Note: ** = $p < 0.01$; *** = $p < 0.001$. → = Paths relationship between constructs of study.

## 6. Discussion and Implication

Our work explored how SMMA impact revisit intention through brand trust and brand loyalty, respectively, and how revisit intention is influenced sequentially through brand trust and brand loyalty. Social media platforms are recognized as a pillar of a business brand's success. Most hospitality businesses (hotels, restaurants, and coffee shops)

rely on these channels to attract more travelers or customers by creating different content on these platforms. However, sequential mediation in the SMMA area has not previously been examined. The relationship between SMMA and branding, such as brand equity and consumer behavior, has been limited in previous studies [7,26,46]. This work investigated the impact of SMMA in the hospitality industry and provided new perspectives for SMMA literature by exploring the unexplained connection between SMMA and intention to revisit.

First, the results of this study confirmed for the first time the sequential mediation: SMMA influences the level of trust among customers who visit the Facebook pages of coffee-shop brands. In turn, this attitude influences brand loyalty, which increases customers' intention to revisit the same location. With effective efforts from coffee shops' Facebook pages, the customers grow their confidence in the brand, which increases the level of brand loyalty and encourages the customer to return to the same place in the future. By classifying the sequential mediation effects in the link between SMMA and revisit intention, this study contributes to the SMM literature's theoretical development that has not provided a clear relationship between SMMA and revisit intention.

Second, by exposing revisit intention in this method, we contribute to the understanding of how Facebook pages can use SMMA to enhance their brand's profitability. The results of this evidence confirmed that SMMA improve and enhance the strength of customers' revisit intention for coffee, because visitors with high involvement on coffee-shop social media pages are more expected to revisit these pages in the future [46]. The finding is consistent with earlier studies that found that social media networks can influence customers [45,86,87]. Therefore, this study determined that customers who share their experiences, opinions about brands, and purchase choices or comments about brands will affect whether other individuals show an interest in the brand or intend to revisit the place. Third, the current study shows a positive relationship between SMMA and brand loyalty. This result is consistent with previous studies that used student samples [8,85] and the luxury-brand sector [25].

Fourth, the outcomes here confirmed that SMMA significantly and positively influence brand trust. These results follow previous studies [6,64] which found that SMMA enhance customer trust. Previously, customers found it difficult to trust brands on social media; however, with the development and increase in users on social media platforms, companies have sought to provide a safe and trustworthy environment for users. Brand trust can be a significant reason for creating a strong and long-lasting relationship between the customer and the company.

Fifth, the findings concerning the mediating effect of brand trust show that brand trust partially mediates the relationship between SMMA and revisit intention. Thus, the consumers who are highly involved with SMMA have proposed an intention to revisit the coffee-shop location, as their trust in the brand increases. We found that brand trust and brand loyalty only partially mediated the effects of SMMA on revisit intention, so that even when brand trust and brand loyalty were concurrently controlled for, SMMA still had a direct and positive relationship with revisit intention. In other words, brand trust and brand loyalty cannot fully explain why SMMA increase revisit intention.

For the first mediated pathway, we noted that in the first illustration, SMMA had a positive influence on brand loyalty by offering effective brand-building potential relationships. Meanwhile, in the second instance, loyalty influenced revisit intention. A theoretical consequence of this finding is that consumers' interactions with brands on social media result in revisit intention via a conceptual mechanism or a chain of effects. SMMA affect brand loyalty via the generation of effective tools, the development of a relationship with customers, and the stimulation of engagement and interaction, which affects revisit intention.

The second path is the brand-trust pathway. We note that SMMA exert a positive impact on revisit intention via brand trust. This result agrees with the other empirical studies in SMM literature. For instance, in the Ibrahim and Aljarah [17] study of customers who followed the Facebook profiles of five-star hotels in Northern Cyprus, it was confirmed

that SMMA affected revisit intention directly and indirectly via brand trust and brand loyalty. On the other hand, the results suggested that brand trust and brand loyalty partially mediate the effect of SMMA on revisit intention. This finding can be attributed to the S-O-R model: stimuli → organism → response [35]. The S-O-R model represents the influence of environmental stimuli (here, SMMA) on the emotional and cognitive reactions of an organism (here, brand trust or brand loyalty), which, in turn, form the behavioral responses (revisit intention) of the organism. This model specifies that the relationship between stimulus, organism, and response is matched with the organism acting as a mediator between stimulus and response [88]. Stemming from our results, the brand trust mediation role explains more about brand loyalty in the relationship between SMMA and revisit intention.

### 6.1. Managerial Implications

The results reveal that brand loyalty and brand trust act as major mediating roles between SMMA and revisit intention. First, this study found that SMMA increase behavioral outcomes (brand loyalty, brand trust, and revisit intention); therefore, SMMA stimulate the revisit intention directly and indirectly by improving brand loyalty and brand trust. For instance, managers and marketers in coffee shops must make special efforts to concentrate on SMMA (entertainment, customization, interaction, WOM, and trendiness) to improve customer responses. SMMA convert customers to promoters who develop, adapt, and exchange relevant information about different brands and their unique products and services [8,57,89]. SMMA, as a stimulus factor, create a secure exchange of relationships between consumers and businesses and strengthen social connections between followers and brands. Therefore, feelings of loyalty to the coffee shop will increase revisits to the same place.

Second, SMM managers are stimulated to frame their SMM efforts to lead to direct revisit intention by engaging brand loyalty or brand trust toward the company. For example, suppose managers find that consumers or users of social media platforms have a low level of trust in the coffee shop's brand or a poor commitment the customer–coffee-shop relationship in the future, they might consider amplifying SMMA. For instance, increasing social media posts containing enjoyable experiences and entertainment options such as photo contests, games, and videos attract the users' attention and increase their enjoyment. Marketers can satisfy customers by directing services, such as customized posts (e.g., Facebook services), toward a specific consumer or consumer group [90]. Coffee shops could provide a more straightforward interaction method by sharing and exchanging content and opinions with users to spread positive WOM. Consequently, the company should improve customer–company relationships by providing timely and trendy information about the brand.

Third, within the hospitality industry, the coffee-shop business is no exception to the precept that customer behavioral outcomes, such as effective interactions with a loyal customer, long-term relationships with trusted customers, and revisit intention, are vital for their worth and success. Our results show that SMMA enhance customer revisit intention via brand loyalty and brand trust, which support coffee shops' brand executives and managers as they plan their marketing strategies. Marketing managers are encouraged to structure their SMM contact efforts, including customer-friendly sharing of fun and interesting content. In that case, the managers may consider improving customers' expectations of SMMA by better communicating the business SMMA and engaging in more interaction on the brand social media page.

### 6.2. Limitations and Future Studies

The presented research has some limitations that can be considered research opportunities. First, we gathered research through a cross-sectional study, and future research should consider involving a longitudinal study overtime to foster an understanding of the influence of SMMA on behavioral outcomes (brand loyalty, brand trust, and revisit inten-

tion). Second, our sample of one field (tourism and hospitality industry, more specifically coffee shops) may have limited our findings' generalizability. Future studies could examine whether different industries (e.g., e-commerce and education) would expose different outcomes and can concentrate on the different types of business within the hospitality industry (e.g., restaurant, hotel, travel agencies, and travel tour companies), with comparisons between industries. The third limitation is that the research was conducted in an emerging country (i.e., Northern Cyprus), considered a Middle Eastern nation. Future research in SMM might compare the differences in the level of globalization among countries (high global, less global) or the level of income for countries or different cultures (Eastern, Western). Fourth, we only focused on one age group (adult students). To further generalize our findings across generations, we recommend exploring the model with different age groups and measuring the effects of generation levels (X.Y.Z) on social media sites.

Fifth, our findings were limited by concentration on only one city's coffee shops with a student clientele. Our sample university students may have failed to characterize broader populations of coffee shops that engaged in social media activities. Selecting nonstudent sample populations in varied backgrounds may enlarge the research scope. Besides those outlines, our study was conducted through questionnaire-based surveys. Future studies can employ different methods to find a deeper understanding of the role of SMM in enhancing the consumer response in the coffee-shop industry, like an experimental study, or a mixture of both quantitative and qualitative methods. Our sixth limitation was that our study investigated only one social media platform (Facebook). Future studies will benefit from researching the differing effects of SMM among multiple social media platforms (e.g., Twitter and Instagram). Further analysis will examine whether individuals from different countries or cultures prefer different social media platforms or not.

Finally, the current study examines the influence of SMMA on behavioral outcomes (brand loyalty, brand trust, and revisit intention), so future research could examine the effects of SMMA on other brand-related outcomes (i.e., online brand community identification, brand attachment, emotional attachment, and intention to follow the advice). The moderator roles (age, gender) have not been assessed in this research for several reasons: the sample in our study encompasses unevenly distributed categories of gender or age; the number of females was almost half the number of males, and the dominant group aged between 18 and 25 years old (80.6%); furthermore, the unbalance and unequal in-sample size each of the groups of the moderator variable leads to the underestimation of the moderating effect [91]; in addition, the power to detect gender or age as a moderator variable is reduced [92]. Future research should examine demographic variables such as (gender, age, employment, income level, education, and personality) as moderator variables in the SMMA context.

**Author Contributions:** Conceptualization, B.I. and A.A.; methodology, B.I. and A.A.; software, A.A.; validation, B.I. and A.A.; formal analysis, A.A.; investigation, B.I.; resources, B.I. and D.S.; data curation, A.A.; writing—original draft preparation, B.I.; writing—review and editing, B.I. and D.S.; visualization, D.S.; supervision, B.I. and A.A.; project administration, A.A. All authors have read and agreed to the published version of the manuscript.

**Funding:** This research received no external funding.

**Institutional Review Board Statement:** Not applicable.

**Informed Consent Statement:** Not applicable.

**Data Availability Statement:** The data presented in this study are available on request from the corresponding author.

**Acknowledgments:** The authors are grateful to all the students of Girne American university, the editors, and the anonymous referees for valuable comments and suggestions.

**Conflicts of Interest:** The authors declare no conflict of interest.

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
