# Peer review of "Linking Social Media Marketing Activities to Revisit Intention through Brand Trust and Brand Loyalty on the Coffee Shop Facebook Pages: Exploring Sequential Mediation Mechanism"

_sustainability, doi:10.3390/su13042277_

Round 1
Reviewer 1 Report
The main problem with this article is the fact that the authors did not demonstrate and not even shown what is the link between their research and sustainability. Even more, sustainability is not even mentioned once in the entire article.
Author Response
Title: Sustainability-1090261- Special Issue "Social Media Influence on Consumer Behaviour"
Comments of Reviewer 1:
Dear Reviewer,
Thank you very much for your time and thoughtful directions regarding our paper. We thoroughly reviewed your comments and extensively revised to address raised issues within our best knowledge (in red color). We hope you are satisfied with how we have addressed your various concerns. Again, we sincerely appreciate your effort and time in reviewing our study and helping us make a better manuscript.
Here is a point-by-point response to the reviewers’ comments and concerns
- The main problem with this article is the fact that the authors did not demonstrate and not even shown what is the link between their research and sustainability. Even more, sustainability is not even mentioned once in the entire article.
Author response:
Thank you for this comment. We do agree with your that our manuscript do not discuss sustainability in any of its section. That is because sustainability is out of the aim of our study. This manuscript has been submitted to the special issue entitled: Social Media Influence on Consumer Behaviour. (please see https://www.mdpi.com/journal/sustainability/special_issues/Social_Media_Influence_Consumer_Behaviour ).
“The main aim of this Special Issue is to deepen our understanding of the changes that the use of social media has brought about both with respect to the consumer and to organizations and companies”. Both of guest editors are interested in the effect onf social media on consumer behavior. They have even not mentioned sustainability, or any related term, in the content of the special issue. The majority of papers that published in that special issue are not related to sustainability in any way. That was the authors’ main motivation behind submitting our manuscript to that special issue, which we believe it fit very well with the aim of the special issue.

Reviewer 2 Report
Summary: The current manuscript explores how social media marketing activities (SMMA) affect brand loyalty, brand trust and revisit intention for coffee shops in Northern Cyprus. Particularly, the authors obtained responses from 415 undergraduate students and applied a structural equation modelling approach. Their main results show that SMMA is a stronger predictor of revisit intention than brand loyalty and brand trust. Overall, this job has important theoretical contribution for the development of business strategies aiming to better characterize consumer decisions through social media activities. Nevertheless, I feel this work needs to work harder in the topics and lines of research to be more suitable to be published in Sustainability. Indeed, there are remarkable issues that limit its contribution to the literature in sustainability in its current form. Broadly speaking, remarkable improvements in methodological procedures, bibliography indications, grammar and discussions are needed to fully evaluate the validity of the results.
Below, my questions are arranged by the order in which they were motivated while reading the manuscript:
1. Abstract: The authors are good storytellers. Nevertheless, it should be better explained some concepts that perhaps are not familiar to the readership of Sustainability, such as “revisit intention”. It is neither shown why the authors selected the Cyprus country.
- Introduction: While I do agree with the research gap the authors outline, they should make a better effort in contextualizing the research gap with more (recent) literature addressing the main authors’ topic. Surprisingly, there is a lack of:
- A clear definition of SMMA or “inner states”, as these concepts may be useful for the readership of Sustainability.
- Important recent literature as the authors do not include any paper dated by 2020.
Zaenab, A., & Sulhaini, H. S. A. (2020). The Effect of Electronic Word of Mouth in Social Media and Experiential Value on Destination Image Dan Revisit Intention after Earthquake in Lombok. Global Journal of Management And Business Research.
Bigne, E., Andreu, L., Perez, C., & Ruiz, C. (2020). Brand love is all around: loyalty behaviour, active and passive social media users. Current Issues in Tourism, 23(13), 1613-1630.
5. You have not even mentioned the word “sustainability” or used papers related to the effects of social media on sustainable business actions. Please kindly read the aim of this journal.
6. Well-developed hypothesis development.
7. Please, improve the quality of all the images (e.g., image 1 has important defects).
8. Mechanical writing issues were distracting throughout and lowered the quality of the entire manuscript. The frequency of these grammatical errors is unacceptable given the quality expectations of this journal. My recommendation is to hire a proofreader to assist you in correcting these grammatical issues. These mistakes are too frequent for me to specifically outline here, and average at least 10-15 mistakes per page of the manuscript.
9. Method: Well done! Nevertheless, please, explain in a clearer way why you have not assessed the moderator effect of gender, age or level of incomes in your analyses. Also it would be a must to explain the implications of using Cyprus as a country of reference, and students as participants’ requirements, as the previous conceptions towards mobile payments may have influenced the final results and being different from other Eastern or Western countries.
10. The authors would need to more deeply explain your contribution to the literature in digital, sustainability research, but also to those studies implemented in the field of marketing research.
11. Highly simplistic and obvious your limitations, managerial implications and further research.
Best of luck as you continue with this research.
Author Response
Title: Sustainability-1090261- Special Issue "Social Media Influence on Consumer Behaviour"
Response to Reviewer #2
Comments of Reviewer
Summary: The current manuscript explores how social media marketing activities (SMMA) affect brand loyalty, brand trust and revisit intention for coffee shops in Northern Cyprus. Particularly, the authors obtained responses from 415 undergraduate students and applied a structural equation modelling approach. Their main results show that SMMA is a stronger predictor of revisit intention than brand loyalty and brand trust. Overall, this job has important theoretical contribution for the development of business strategies aiming to better characterize consumer decisions through social media activities. Nevertheless, I feel this work needs to work harder in the topics and lines of research to be more suitable to be published in Sustainability. Indeed, there are remarkable issues that limit its contribution to the Literature in sustainability in its current form. Broadly speaking, remarkable improvements in methodological procedures, bibliography indications, grammar and discussions are needed to fully evaluate the validity of the results.
Below, my questions are arranged by the order in which they were motivated while reading the manuscript:
- Abstract: The authors are good storytellers. Nevertheless, it should be better explained some concepts that perhaps are not familiar to the readership of Sustainability, such as "revisit intention". It is neither shown why the authors selected the Cyprus country.
Responses from Authors:
- We appreciate your encouraging "The authors are good storytellers" judgment. Per your insightful advice, we have provided a clear explanation of the following concepts “revisit intention” in abstract, Literature review and introduction sections. Please see Page: 1 Line: 14 to 15.
Please see Page: 2 Line: 60 to 62.
Please see Page: 4 Line: 168 to 170.
- For second points. We appreciate raising this issue. We have tried to address your insightful critique. We have updated the arguments about " why the authors selected the Cyprus country". Please see Page: 3 Line: 100 to 105.
Comments of Reviewer 2
- Introduction: While I do agree with the research gap the authors outline, they should make a better effort in contextualizing the research gap with more (recent) Literature addressing the main authors' topic. Surprisingly, there is a lack of:
- Important recent Literature as the authors do not include any paper dated by 2020.
Responses from Authors
- Thank you for pointing this out. We agree with this comment and thank you for pointing this out. Some of the references for the research gap were not recent. Therefore, we revamped the literature and research gaps. We have checked and modified the research gap section and Literature Review section with recent Literature as the following :
- Unpacking the relationship between social media marketing and brand equity: The mediating role of consumers' benefts and experience (2020)
- Does E-Brand experience matter in the consumer market? (2020)
- Explaining the impact of social media marketing activities on consumer-based brand equity and love
- Do Social Media Marketing Activities Enhance Consumer Perception of Brands? A Meta-Analytic Examination (2020)
- Perceived Social Media Marketing Activities and Consumer-Based Brand Equity: Testing a Moderated Mediation Model (2020)
- Driving COBRAs: the power of social media marketing (2020)
- Social Media Activities and its Influence on Customer‐Brand Relationship: An Empirical Study of Apparel Retailers' Activity in India (2021)
- Customer Behavior as an Outcome of Social Media Marketing: The Role of Social Media Marketing Activity and Customer Experience (2020)
- Investigating the role of social media marketing on value co-creation and engagement: An empirical study in China and Hong Kong (2021)
- The dynamic stimulus of social media marketing on purchase intention of Indonesian airline products and services (2020)
- The influence of perceived social media marketing elements on consumer–brand engagement and brand knowledge (2020)
- Zaenab, A., & Sulhaini, H. S. A. (2020). The Effect of Electronic Word of Mouth in Social Media and Experiential Value on Destination Image Dan Revisit Intention after Earthquake in Lombok. Global Journal of Management And Business Research.
- Bigne, E., Andreu, L., Perez, C., & Ruiz, C. (2020). Brand love is all around: loyalty behaviour, active and passive social media users. Current Issues in Tourism, 23(13), 1613-1630.
- The Role of Trust in Understanding the Impact of Social Media Marketing on Brand Equity and Brand Loyalty (2019)
- How corporate social responsibility contributes to strengthening brand loyalty, hotel positioning and intention to revisit? (2020)
Comments of Reviewer 2
- A clear definition of SMMA or "inner states", as these concepts may be useful for the readership of Sustainability
Responses from Authors
- We agree with this comment and thank you for pointing this out. We have worked hard to provide a clear definition of SMMA or "inner states" in the introduction ,Literature review and Theoretical Background sections.
- SMMA - Please see Page: 2 Line:55 to 57. Please see Page: 4 Line:161 to 167.
- Inner states- Please see Page: 3 Line:142 to 150.
Comments of Reviewer 2
- You have not even mentioned the word "sustainability" or used papers related to the effects of social media on sustainable business actions. Please kindly read the aim of this journal.
Responses from Authors
Thank you for this comment. We do agree with your that our manuscript do not discuss sustainability in any of its section. That is because sustainability is out of the aim of our study. This manuscript has been submitted to the special issue entitled: Social Media Influence on Consumer Behaviour. (please see https://www.mdpi.com/journal/sustainability/special_issues/Social_Media_Influence_Consumer_Behaviour ).
"The main aim of this Special Issue is to deepen our understanding of the changes that the use of social media has brought about both with respect to the consumer and to organizations and companies". Both of guest editors are interested in the effect onf social media on consumer behavior. They have even not mentioned sustainability, or any related term, in the content of the special issue. The majority of papers that published in that special issue are not related to sustainability in any way. That was the authors' main motivation behind submitting our manuscript to that special issue, which we believe it fit very well with the aim of the special issue.
Comments of Reviewer 2
- Well-developed hypothesis development.
Responses from Authors
- We appreciate your encouraging " Well-developed hypothesis development." judgment. We sincerely appreciate your effort in reviewing our study and helping us make a better manuscript
Comments of Reviewer 2
- Please, improve the quality of all the images (e.g., image 1 has important defects).
Responses from Authors
- Thank you for pointing this out. We agree with this comment. Therefore, we have improved the quality of all the manuscript images.
Comments of Reviewer 2
- Mechanical writing issues were distracting throughout and lowered the quality of the entire manuscript. The frequency of these grammatical errors is unacceptable given the quality expectations of this journal. My recommendation is to hire a proofreader to assist you in correcting these grammatical issues. These mistakes are too frequent for me to specifically outline here, and average at least 10-15 mistakes per page of the manuscript.
Responses from Authors
- We agree with the sentiment expressed here. We have revamped the paper's language substantially. We have reviewed and made changes in the entire document, please check the red colour changes.
Comments of Reviewer 2
- Method: Well done! Nevertheless, please, explain in a clearer way why you have not assessed the moderator effect of gender, age or level of incomes in your analyses. Also it would be a must to explain the implications of using Cyprus as a country of reference, and students as participants' requirements, as the previous conceptions towards mobile payments may have influenced the final results and being different from other Eastern or Western countries
Responses from Authors
We appreciate your encouraging "Well done" judgement. We appreciate raising this point out. We have tried to address your insightful critique (moderator effect) in the limitation section in two reasons: first, the sample in our study encompasses unevenly distributed categories of gender or age; the number of females was almost half the number and the dominant group aged between 18 and 25 years old (80.6%); furthermore, the unbalance and unequal in sample size each of the groups of the moderator variable will lead to the underestimation of the moderating effect [93]; also the power to detect gender or age as a moderator variable is reduced [94] , Please see Page: 12 Line: 495 to the end of the page. Second, In the limitation section, we have acknowledged that our study did not use moderator effect in the relationship between construct of study and we suggest to explore the different type of moderator (gender, age or level of incomes) in future studies. Please see Page: 12 Line: 495 to the end of the page.
For second points, related use Cyprus as reference and student as participants. We have worked harder to point out the previous points, and shown the reason to use Cyprus as reference and student as participants. The following line in the method and introduction section deals with this point in this version.
Please see Page: 3 Line: 100 to 105.
Please see Page: 7 Line: 277 to 282.
Comments of Reviewer 2
- The authors would need to more deeply explain your contribution to the Literature in digital, sustainability research, but also to those studies implemented in the field of marketing research.
Responses from Authors
We appreciate raising this issue. Our study's contributions focused on marketing research in general (specifically digital research) and hospitality research. We don't think we have a contribution to sustainability research. We have several motivations and contributions to perform this research as followings: This work contributes to SMMA literature in the tourism and hospitality industry and digital marketing area in several ways.
- This first gap in Literature paints an incomplete picture of the SMMA– customer responses relationship and limits our understanding. Our research is one of the few studies that examine the purpose, brand trust, brand loyalty, revisit intention, and SMMA together in the tourism and hospitality industry.
- A second gap in the Literature concerns the underlying mechanism of how SMMA influence the tourism and hospitality business and the customers who follow the Facebook pages of coffee shops. Therefore, the current study fills this research gap by empirically examining SMMA’s role in promoting brand trust, brand loyalty, and revisit intention in the top two franchise coffee companies in Northern Cyprus.
- The third research gap concerns the conditions under which SMMA might enhance revisit intention in the hospitality context (specifically coffee shops). No current research examines the mediating roles of other aspects of cognitive and emotional states such as brand trust and brand loyalty in the relationship between SMMA and revisit intention. By examining the relationship between SMMA and revisit intention considering a brand trust and brand loyalty as a mediating factor, this research bridges this gap in the SMMA literature and calls for further investigation into different mediation roles between SMMA and consumer responses.
- The fourth research gap covers the sequential mediation effects of brand loyalty and brand trust in the relationship between SMMA and revisit intention. SMMA intend to influence revisit intention through the mediation effect of brand trust and brand loyalty.
Comments of Reviewer 2
- Highly simplistic and obvious your limitations, managerial implications and further research.
Responses from Authors
As per your sensible advice, we strengthened the following sections: limitations, managerial implications and further research. We have tried to address your sensible suggestions to the best of our knowledge.
Please see Page: 11 Line: 426 to 461.
Please see Page: 12 Line: 462 to 504.

Round 2
Reviewer 1 Report
The paper is well written with lots of research made on this topic and has some improvements. The way through which the authors debate this topic is very good. I can see that the authors worked a lot in accumulating knowledge on this field and put an accent on their personal opinions regarding the field of social media marketing.
The paper has 2 figures and 3 tables, that can help the readers to understand the topic better and also the sequential mediation model is well developed and used.
The authors interpret the findings bearing in mind also the points of view of the results obtained in his research but also past studies carried on this special topic.
The bibliography consists of 94 cited books and scientific articles, but the newest references are from 2021, the rest being from the early years going up to 1973.
Reviewer 2 Report
The authors have properly addressed my remaining concerns!